# E-BDL: Enhanced Band-Dependent Learning Framework for Augmented Radar Sensing

**DOI:** 10.3390/s24144620

**Published:** 2024-07-17

**Authors:** Fulin Cai, Teresa Wu, Fleming Y. M. Lure

**Affiliations:** 1School of Computing and Augmented Intelligence, Arizona State University, Tempe, AZ 85287, USA; fulin.cai@asu.edu; 2ASU-Mayo Center for Innovative Imaging, Arizona State University, Tempe, AZ 85287, USA; 3MS Technologies Corporation, Rockville, MD 20580, USA; fleming.lure@mstechnologies.com

**Keywords:** radar sensing, spectrogram, sub-band, deep learning, contrastive learning

## Abstract

Radar sensors, leveraging the Doppler effect, enable the nonintrusive capture of kinetic and physiological motions while preserving privacy. Deep learning (DL) facilitates radar sensing for healthcare applications such as gait recognition and vital-sign measurement. However, band-dependent patterns, indicating variations in patterns and power scales associated with frequencies in time–frequency representation (TFR), challenge radar sensing applications using DL. Frequency-dependent characteristics and features with lower power scales may be overlooked during representation learning. This paper proposes an Enhanced Band-Dependent Learning framework (E-BDL) comprising an adaptive sub-band filtering module, a representation learning module, and a sub-view contrastive module to fully detect band-dependent features in sub-frequency bands and leverage them for classification. Experimental validation is conducted on two radar datasets, including gait abnormality recognition for Alzheimer’s disease (AD) and AD-related dementia (ADRD) risk evaluation and vital-sign monitoring for hemodynamics scenario classification. For hemodynamics scenario classification, E-BDL-ResNet achieves competitive performance in overall accuracy and class-wise evaluations compared to recent methods. For ADRD risk evaluation, the results demonstrate E-BDL-ResNet’s superior performance across all candidate models, highlighting its potential as a clinical tool. E-BDL effectively detects salient sub-bands in TFRs, enhancing representation learning and improving the performance and interpretability of DL-based models.

## 1. Introduction

Utilizing the Doppler effect, which manifests itself as frequency or phase shifts in returned signals, radar sensors can remotely capture kinetic and physiological motions of individuals while maintaining privacy preservation [1,2,3]. Kinetic motions are presented in the micro-Doppler frequency fluctuations resulting from movements of body joints, encompassing vibrations or rotations alongside straight-line motion [1]. Similarly, subtle chest expansion induced by physiological motions (breathing and heartbeat) is captured by analyzing changes in both frequency and phase of radar signals [2,3]. The returned radar signal is expected to contain adequate information about an individual’s body movements and vital signs.

Artificial intelligence (AI) can extract salient information from radar signals, enabling radar sensing to be applied in various healthcare domains, such as gait recognition, fall detection, and vital-sign measurement [4,5,6]. Traditional AI approaches rely on feature engineering to extract task-specific features from radar signals, followed by machine learning (ML) models to correlate extracted features with ground truth labels. Unlike traditional ML, deep learning (DL) models the relationship between inputs and outputs while learning to identify effective features in an end-to-end manner [7]. Additionally, DL models have shown superiority in pattern learning and flexibility in adapting to various applications over traditional ML models when applied to radar sensing data [5,6].

Representation schemes for radar data play a fundamental role in the overall performance of DL-based frameworks [6,8,9]. One of the common schemes is time–frequency representation (TFR), particularly spectrograms, which represent a signal in both time and frequency domains. They can be extracted from the return signals of any continuous wave radar, which is one of the simplest forms of radar systems using short-time Fourier transform (STFT) [1]. TFR is treated as a 1-channel image in a DL-based framework. DL-based models based on 2D convolution neural network (CNN) have had great success in recognition or classification tasks of radar sensing applications such as activity recognition [6], gait classification [10], and hemodynamics scenarios classification [11].

However, current research in radar sensing applications utilizing TFR as the input has not fully explored characteristics of band-dependent patterns: motion patterns vary across frequency bands that depend on speeds of targets, and discernible differences in power scales of patterns that depend on radar cross-sections (RCS) of targets. Using human walking as an example, body joints such as hips, knees, and ankles move within various ranges of angles and angular velocities, respectively [12]. Examples of micro-Doppler signatures for human walking in [13] show that signatures indicating knee movements occur below 200 Hz, while those indicating ankle and toe movements span the [0, 400] Hz range. Additionally, the torso and hips move more slowly than the legs and feet. As seen in [10], radar reflections from the torso and hips appear in low-frequency regions (<300 Hz), while reflections from the legs and feet can reach higher-frequency regions (>300 Hz). Therefore, detected motion patterns of body joints vary and fall into different frequency bands. RCS varies with the target’s size, geometry, material, transmitter frequency, polarization, and aspect angles relative to the radar transmitter and receiver, affecting power scales [1]. Under specific radar equipment, RCS is associated with the properties of the target, so larger body parts (e.g., torso) with higher RCS reflect significantly higher power in TFR, while smaller body parts (e.g., feet) reflect relatively lower power. For the detection of vital signs, compared to the human heartbeat (within 1–2.5 Hz), breathing occurs at lower frequencies (within 0.1–0.9 Hz). It is characterized by more pronounced chest expansion, indicating a larger RCS with higher power values [2,6].

The band-dependent patterns challenge DL models using TFR as input, particularly CNNs. The variations of motion patterns conflict with the translation invariance property, which is essential for CNN-based models [14]. Thus, CNN-based models may overlook characteristics that depend on frequency in TFR. Additionally, bias in feature extraction towards regions with high power scales leads to the loss of information from regions with low power scales [15]. To address these issues, research in the audio field, also preferring to use TFR as input, either limits kernel sharing of the convolution layer [16] or employs independent convolution and normalization layers for each predivided sub-band [14,17]. However, the design of kernel sharing and division of sub-bands are heuristic or require trial-and-error efforts. To divide sub-bands automatically, an adaptive filtering method incorporating low-, band-, and high-pass filters effectively separates sub-bands by introducing trainable cut-off frequencies and damping ratios into a CNN-based framework [15]. However, this method may fragment signals from specific body parts (e.g., feet), which span across entire frequency ranges, into three parts, and it overlooks the interaction among visualized features in sub-bands for CNN. In addition, these works utilize parallel convolution models or layers whose parameters are not shared to extract feature representation for sub-bands, thereby not fully leveraging the capacity and generalization of CNNs.

In radar sensing applications, research utilizing physiological motions mainly relies on band-pass filtering approaches to decompose signals into sub-bands that reflect heartbeat and breathing, respectively [6,11]. The ranges of passbands are empirically determined and may vary from case to case. Regarding kinetic motions, a band-dependent learning framework (BDL) is developed for abnormal gait recognition for Alzheimer’s disease (AD) and AD-related dementia (ADRD) risk evaluation employed [10]. BDL uniformly divides a spectrogram containing gait signatures into multiple sub-bands with equal sizes for representation learning and normalization. This approach mitigates the issue of scale differences between sub-bands and avoids the loss of information in regions with lower power values. However, the uniform division inevitably disperses gait features into multiple sub-bands, making it challenging for DL models to capture the complete information.

To adaptively identify salient sub-band regions and fully leverage band-specific features, this study introduces a novel DL-based framework called the Enhanced Band-Dependent Learning framework (E-BDL). This framework integrates adaptive sub-band filtering, representation learning, and sub-view contrastive techniques in an end-to-end manner. This research makes three primary contributions:E-BDL, leveraging band-dependent patterns, is proposed to enhance the recognition ability of the DL-based models for radar sensing applications by addressing the challenges from variations in patterns and differences in power scale associated with sub-bands.An adaptive sub-band filtering module identifies salient sub-bands with band-dependent patterns as sub-views. A representation module extracts and normalizes features within each sub-view. A sub-view contrastive loss function supports both modules in exploring distinct sub-view spaces and ensuring discrimination.The proposed model’s performance is validated using two radar-sensing datasets: one for gait abnormality detection in ADRD risk evaluation and another for vital-sign monitoring in hemodynamics scenario classification. Experimental results demonstrate the superior performance of E-BDL-ResNet and the interpretability of E-BDL through learned sub-band filters.

This paper is organized as follows. Section 2 provides the details of the proposed framework, followed by the experiments in Section 3. The discussion and the conclusions are presented in Section 4 and Section 5, respectively.

## 2. Proposed Method

An illustration of E-BDL is shown in Figure 1. A TFR $x\in R^{T\times Fs}$ serves as the input to the E-BDL following three modules: (1) An adaptive sub-band filtering module (see Section 2.1): *M* sub-band filters denoted as $\left\{b_{i}\in R^{1\times Fs}|i=1,2,\ldots ,M\right\}$ are generated and applied in parallel to *x* in parallel to produce *M* sub-views, $\left\{x_{i}^{\prime }\in R^{T\times Fs}|i=1,2,\ldots ,M\right\}$; (2) A representation learning module: each $x_{i}^{\prime }$ is transformed and normalized into a *L*-dimensional representation vector, $∥z_{i}∥$, by a DL-based encoder (see Section 2.2); and (3) A sub-view contrastive module: class proxies of each sub-view are initialized in representation space, and a contrastive-based loss function is employed to adjust the metric distance relationships between $∥z_{i}∥$ and corresponding proxies for filter learning and representation space construction (see Section 2.3).

### 2.1. Adaptive Sub-Band Filtering Module

Features in TFR have exhibited band-dependent patterns, including variations in patterns and power scales associated with frequencies. An adaptive sub-band filtering module is developed to identify salient sub-bands from a vision perspective by scaling power values at each frequency. The generalized Gaussian distribution [18] is modified to create a scaling factor for each frequency bin. Frequency bins ranging in [−Fs/2,Fs/2] are linearly normalized to [−1,1]. Given a frequency bin, denoted as fb, the scaling factor is determined by
(1)$$g\left(fb;\mu ,\sigma ,\beta \right)=\exp\left(-{\left(\frac{\left|fb-\mu \right|}{\sigma }\right)}^{\beta }\right)$$
where μ, σ, and β represent parameters of location, scale, and shape, respectively. A general example is displayed in Figure 2a, where μ and σ are initialized to be 0 and 1, respectively. The scaling factors of the generated filter remain 1 in the salient sub-band (top) but decay in the adjacent regions (sides).

In practice, the top part of the filter is determined by μ and σ. In other words, μ and σ control the center frequency and the bandwidth of the filter, respectively. β determines the slope of the side parts, controlling the transition of the band edge. Parameters μ and σ are trainable parameters. As a preliminary study, the β is set to 8 to ensure gradient quality from the sides and convergence speed of the model. For the model initialization, μ of filters are uniformly distributed along the frequency axis according to the number of filters while σ of filters are set to 1 (see Figure 2b for an initialization example using three filters). The *i*th sub-band filter is $b_{i}=g\left(fb;\mu_{i},\sigma_{i},\beta \right)\in R^{1\times Fs}$ and the *i*th sub-view $x_{i}^{\prime }$ is determined by $x\times b_{i}$.

### 2.2. Representation Learning Module

Each $x_{i}^{\prime }$ represents a salient and unique sub-view filtered out from the input TFR. DL models commonly treat TFR as a 1-channel image and transform TFR information into a discriminative representation vector [5,6]. In this module, *M* sub-views are transformed into *L*-dimensional representation vectors by the same encoders, and then the representation vectors are normalized onto a hypersphere space by $L_{2}$-normalization. Consequently, $\left\{∥z_{i}∥\in R^{1\times L}|i=1,2,\ldots ,M\right\}$ is obtained. Please note that using the same encoder ensures that sub-views are transformed into the same representation space to achieve fair contrast in the next module.

In this study, the backbone of the pretrained ResNet-18 [19] is employed for representation learning to leverage its generalization capabilities on edge and shape detection. The feature map output from the backbone is input to a projection head layer, comprising a 512×512 linear layer followed by batch normalization, ReLU activation, and another 512×128 linear layer for feature refinement and compression. The combination of the backbone and the projection layer is utilized as the encoder. To fulfill the input requirement of the encoder (3-channel), the 1-channel spectrogram is replicated three times. During training, gradients are propagated backward from the subsequent modules to update the parameters of the encoder. The proposed approach is model-agnostic, since the encoder can be replaced by any DL-based model.

### 2.3. Sub-View Contrastive Module

Data samples from radar sensing applications often exhibit low inter- and intraclass differences, which poses challenges for the generalization of traditional approaches employing cross-entropy loss [20,21]. Cross-entropy loss, designed to match the probability distributions of ground truths, does not account for interrelationships among representation vectors. An effective alternative is the use of metric-based loss functions, which adjust metric relationships among representation vectors to increase interclass variability and intraclass similarity [22]. Additionally, metric-based loss functions can be categorized into sample-to-sample comparison and sample-to-proxy comparison [23,24]. The former methods can capture local relations within a mini-batch but are computationally intensive and require sufficient data for generalization. Instead, the sample-to-proxy comparison strategy associates each sample with proxies of classes, requiring fewer data. In addition, globally learned proxies may be more robust against noisy samples and outliers [23,24].

Data scarcity and model generalization challenges DL-based radar sensing applications [8,9]. Therefore, using the sample-to-proxy comparison strategy, a sub-view contrastive loss function ($SC_{Loss}$) based on metric distance is developed. As seen in Figure 1, the transformed representation vector of each sub-view is assigned to adjust metric relationships among sub-views and proxies in sub-view clusters. Each sub-view cluster contains proxies that correspond to classes. $SC_{Loss}$ has two primary objectives: (1) generating a discriminative representation space for each sub-view, and (2) ensuring that sub-views capture different valuable features.

The proxies are normalized and denoted as $\{∥p_{i,c}∥\in R^{1\times L}|i=1,2,\ldots ,M\;\mathrm{and}\;$c=1,2,…,C}, where *C* is the number of classes. Regarding each $∥z_{i}∥$, the proxies can be divided into four groups: same-view-same-class, $\tilde{p}$, same-view-different-classes, A(i), different-views-same-class, B(i), and others. For each *x* in a mini-batch, $\left\{∥z_{i}∥\right\}$ is obtained from the representation learning module, and the calculation of $SC_{Loss}$ takes the following form: (2)$$SC_{Loss}\left(\{∥z_{i}∥\}\right)=\frac{-1}{M}\sum_{i=1}^{M}\log\left(\frac{e^{\tilde{∥p∥}\cdot ∥z_{i}∥}}{e^{\tilde{∥p∥}\cdot ∥z_{i}∥}+\sum_{p\in A(i)}e^{∥p∥\cdot ∥z_{i}∥}+\sum_{p\in B(i)}e^{\mathrm{ReLU}(∥p∥\cdot ∥z_{i}∥)}}\right)$$
where ReLU(x)=max(0,x). Utilizing the cosine similarity metric, this loss function encourages $∥z_{i}∥$ to be closer to the proxy from the same view and class, while increasing the distance to the proxies from the same view but different classes. To ensure that sub-views capture distinguishable features, the similarity between $∥z_{i}∥$ and the proxies from the same class but different views should be less than 0. The final loss value used for backpropagation is the mean of the $SC_{Loss}$ values of all *x* in a mini-batch. During testing, $\left\{∥z_{i}∥\right\}$ are compared to proxies in $\left\{∥p_{i,c}∥\right\}$ belonging to the same sub-view. Because discriminative features might only occur in a specific sub-view, the class of the closest proxy—indicating the minimum distance from the proxy to the sample—among all comparison results is adopted as the prediction.

## 3. Experiments and Results

### 3.1. Datasets

Two datasets employing radar sensors are utilized: one for vital sign monitoring in hemodynamics scenarios classification and another for gait detection in ADRD risk evaluation. Details of the datasets and evaluation strategies are provided below.

**Vital Sign**: Radar is able to accurately monitor the physiological condition of subjects with contact-free benefit. A 24 GHz continuous wave radar system based on Six-Port technology is implemented to record cardiorespiratory activities, and this publicly available dataset is collected by physicians at the Department of Palliative Medicine at the university hospital Erlangen [25]. Five hemodynamic scenarios are established for 30 healthy volunteers: resting, Valsalva, apnea, tilt-up, and tilt-down. These scenarios are expected to result in various characteristics in cardiorespiratory activities of subjects. The radar focuses on chest movements while subjects undergo each scenario. The captured respiratory, heartbeat, and pulse activities can be derived from the returned signals to understand characteristics of respiratory or cardiovascular activity. In this study, raw radar signals are employed to verify the effectiveness of the proposed approach in detecting these activities.**Gait**: Human gait is an effective biomarker associated with the ADRD risk level, and gait motions during walking can be detected by radar. Gait motions contain micro-Doppler signatures from several body parts, including the torso, legs, feet, etc. These signatures are characterized by different frequency bands, central frequencies, bandwidths, and damp ratios. A large, real-world, publicly available gait dataset for ADRD risk evaluation is lacking. Walking animations are simulated, representing four ADRD levels: normal walk, subtle abnormality walk, moderate abnormality walk, and severe abnormality walk. A total of 8000 animations are generated, with 2000 walking animations for each category, and these animations are presented on a 10 m simulated walking path. Gait motions during walking are monitored by a frequency-modulated continuous-wave radar. The radar detects approximately a 4 s walking period. The returned signals are transformed into micro-Doppler signatures using STFT. The gait signatures representing gait motions are included in a spectrogram. More details can be found in [10].

For the vital-sign dataset, a 5-fold cross-validation strategy, with 80% of the samples for training and the remaining 20% for testing each time, was conducted in [11]. This 5-fold cross-validation strategy is subject-dependent, meaning that samples in the training set and test set could be from the same subject. To identify the model’s generalization across different subjects, the leave-one-out (LOO) strategy is utilized. In this strategy, one subject’s data are used for testing, and the other 29 subjects are used for training each time, ensuring a subject-independent manner. Both validation strategies iteratively cover all samples for testing, with 20% of the data in the training set used as the validation set. Following the evaluation strategy in [10], 8000 samples in the gait dataset are randomly divided into a training set, a validation set, and a test set in a ratio of 7:1.5:1.5. Thus, 1200 samples are allocated to the test set.

### 3.2. Data Preprocessing

Referring to the preprocessing steps described in [11], valid radar recordings from the vital-sign dataset are extracted and downsampled to 100 Hz. The recordings are then segmented into 20 s intervals with a 50% overlap, resulting in each segment containing 2000 complex data points with in-phase and quadrature components. Each segment is filtered using a 4th-order Butterworth band-pass filter with cutoff frequencies of [0.1, 20] Hz. To adapt to the input of CNN-based models, each filtered radar segment is transformed into a two-sided complex spectrogram, which is considered as a time–frequency image, using STFT with a configuration that includes 512 DFT and a Hanning window of 128 points with a 108-sample overlap between windows. The magnitude and phase parts of a spectrogram are extracted separately. The magnitude part is scaled by $g\left(x\right)=10\times \log10\left(\left|x\right|+10^{-5}\right)$. After standardization, an input sample is obtained by combining half of the magnitude part and half of the phase part along the frequency axis. As summarized in Table 1, the samples in the vital-sign dataset are severely imbalanced, with the resting category containing more samples than the other categories. To address this issue, a data augmentation approach is developed in which two samples are randomly selected from the resting class and each of the other four classes. The approach then combines these selected samples in equal proportions to generate a new augmented sample. The label assigned to this augmented sample corresponds to the class from which the nonresting sample is chosen. It should be noted that the data augmentation method is active only during training.

For the gait dataset, following [10], an STFT configuration utilizes 512 discrete Fourier transform (DFT) points, and a Kaiser window of 512 points with a 50% overlap between windows is applied to the reflected radar signals. Dimensions of a spectrogram sample are 512 and 776 for the frequency and time domains, respectively. Spectrograms are scaled by $f\left(x\right)=20\times \log_{10}\left(\left|x\right|+1\times 10^{-5}\right)$ and then standardized. Consequently, a total of 8000 spectrograms are obtained, with 2000 spectrograms for each class (see Table 1).

### 3.3. Experimental Setup

To assess the performance of the proposed approach, ResNet-18 [19], BDL-ResNet [19], EfficientNetV2-S [26], and ConvNeXt-T [27] are employed for comparison. EfficientNetV2-S and ConvNeXt-T are two of the recent state-of-the-art CNN models. As seen in Table 2, these two models contain approximately twice as many parameters and require more inference time for both datasets compared to the other three ResNet-based models. In this study, for simplicity, E-BDL uses the backbone of ResNet-18 as the representation learning module. ResNet-18 is used for the ablation study to demonstrate the advantages of E-BDL. All DL-based models utilize pretrained parameters to leverage their knowledge from image classification tasks. For the vital-sign dataset, a feature-engineering-based benchmark method, ANN [11], is also included in the 5-fold validation for comparison.

For the vital-sign dataset, the batch size for all DL-based models is 32. The learning rates are set to 0.0001, and 0.25 is used for the parameters of the sub-band filters to achieve faster convergence. The models are trained for up to 15 epochs using the Adam optimizer [28]. The number of filters, a hyperparameter ranging from 1 to 6, is determined based on the validation results. For the gait dataset, the learning rate is set to 0.001, and 0.25 is employed for the parameters of sub-band filters. The batch size is set to 32. The seed number for initialization and loading data is fixed at 1024. For achieving comprehensive evaluations, accuracy (Acc) is adopted for the overall assessment, while precision (Pre) and sensitivity (Sen) are used for class-wise evaluation. Due to the significant data imbalance in the vital-sign dataset, balanced accuracy is employed for fair comparison. All experiments are conducted on the Sol supercomputer [29] using the PyTorch 2.0.1 framework [30].

### 3.4. Experimental Results

The 5-fold and LOO validations are conducted for the vital-sign dataset. The evaluation results are summarized in Table 3 for the 5-fold validation and Table 4 for the LOO validation. For the gait dataset, the evaluation results on the test set are presented in Table 5. It is important to mention that the 5-fold validation for the vital-sign dataset is subject-dependent while the other two evaluations are subject-independent.

#### 3.4.1. Results on the Vital-Sign Dataset in 5-Fold Validation

The proposed method, E-BDL-ResNet, achieves an overall accuracy of 0.959, which is slightly higher than ResNet-18 (0.956) and EfficientNetV2-S (0.955), but marginally lower than ConvNeXt-T (0.969). This indicates that E-BDL-ResNet is highly effective and competitive among the candidates across all classes. For the resting category, E-BDL-ResNet achieves the highest sensitivity of 0.967 and competitive precision (0.970), though slightly below ConvNeXt-T (0.973). This verifies the robustness of E-BDL-ResNet in detecting true positives for the resting category. While ANN shows perfect performance in precision (0.993) and sensitivity (1.000) for the Valsalva category, E-BDL-ResNet still performs commendably, with a sensitivity of 0.963 and a precision of 0.943. Similarly, for the apnea category, ANN achieves the highest precision (0.984) and sensitivity (1.000). The few samples in these two categories challenge the DL-based models, but the DL-based models still exhibit competitive precision and sensitivity for the Valsalva and apnea categories while maintaining balanced performance across different categories. For the tilt-up category, E-BDL-ResNet exhibits excellent performance with a precision of 0.984 and a sensitivity of 0.992. These metrics are close to the best-performing ConvNeXt-T (0.985 and 0.997, respectively), showcasing E-BDL-ResNet’s capability in this class. E-BDL-ResNet achieves the highest precision of 0.959 among all models. Its sensitivity is also high at 0.959, though marginally lower than ConvNeXt-T (0.970). This highlights E-BDL-ResNet’s balanced and reliable performance for this category.

While ConvNeXt-T achieves the highest overall accuracy and excels in several class-wise metrics, E-BDL-ResNet exhibits comparable accuracy and is competitive across all classes using fewer parameters and less inference time (See Table 2). Additionally, compared to ResNet-18, E-BDL-ResNet has demonstrated improvements in overall accuracy and its superiority in the resting, tilt-up, and tilt-down categories regarding precision and sensitivity, indicating the benefit of E-BDL. In practice, the resting, tilt-up, and tilt-down categories exhibit the most similar pattern, highlighting the superiority of E-BDL in capturing subtle differences.

#### 3.4.2. Results on the Vital-Sign Dataset in LOO Validation

In addition to the 5-fold validation, where the DL-based models have shown excellent performance, the LOO validation results are presented in Table 4. This validation is deployed in a subject-independent manner to assess the models’ generalization across various subjects. Significant declines are observed in all evaluation metrics in LOO validation compared to the results in the 5-fold validation, attributed to the variance in the subjects of the dataset.

E-BDL-ResNet achieves the highest overall accuracy (0.722) among the evaluated models, slightly surpassing ConvNeXt-T (0.719) and EfficientNetV2-S (0.714). This demonstrates the superior capability of E-BDL-ResNet in generalizing across the entire dataset when using the LOO validation. The performance of all models is drastically degraded in identifying the resting and tilt-down samples. E-BDL-ResNet leads with a precision of 0.567 and a sensitivity of 0.565 in the resting and tilt-down categories, respectively, while EfficientNetV2-S achieves the highest sensitivity (0.502) and precision (0.476) in the resting and tilt-down categories, respectively. This suggests that E-BDL-ResNet is competitive in tackling the challenges caused by the variations of heartbeat and respiration patterns among individuals. ConvNeXt-T leads in performance in both the Valsalva and apnea classes. For Valsalva, ConvNeXt-T achieves the highest precision (0.893) and sensitivity (0.879), closely followed by E-BDL-ResNet with a precision and sensitivity of 0.888 and 0.867, respectively. In the apnea class, ConvNeXt-T outperforms other models with precision (0.852) and sensitivity (0.894). E-BDL-ResNet shows competitive performance with precision (0.825) and sensitivity (0.854), but ConvNeXt-T maintains a marginal edge. For the tilt-up category, E-BDL-ResNet excels with the highest precision (0.874), outperforming ConvNeXt-T (0.838) and EfficientNetV2-S (0.805), and a competitive sensitivity (0.885). These results demonstrate that E-BDL-ResNet is a highly competitive model compared to the two latest state-of-art (SOTA) CNN-based models.

Compared to ResNet-18, E-BDL-ResNet achieves a 1.5% increase in accuracy relative to ResNet, larger than that observed in 5-fold validation. For the class-wise evaluation, E-BDL-ResNet improves precision and sensitivity in the resting, Valsalva, and tilt-up categories while maintaining comparable results in the apnea and tilt-down categories. Particularly, E-BDL-ResNet significantly improves precision by 2.1% and sensitivity by 1.3% in the tilt-up category. Although the limited number of subjects challenges the model’s generalization, E-BDL still demonstrates its ability to improve ResNet’s performance as SOTA when handling heavy subject variabilities.

#### 3.4.3. Results on the Gait Dataset

The gait data include radar reflections from all body joints across various frequency ranges. Compared to the vital-sign data, which mainly relate to the heartbeat and respiration, more complex band-dependent patterns are observed in the gait data. As shown in Table 5, E-BDL-ResNet outperforms all candidates in terms of overall accuracy, achieving 0.943, while EfficientNetV2-S achieves 0.900, ConvNeXt-T 0.917, BDL-ResNet 0.923, and ResNet 0.884. This highlights the superior performance and generalization of E-BDL-ResNet in ADRD risk level evaluation using gait data. Compared to ResNet-18, E-BDL-ResNet significantly improves accuracy by 5.9%. Additionally, E-BDL-ResNet surpasses BDL-ResNet, which uniformly divides frequency bands, by achieving a 2% increase in accuracy through the adaptive identification of salient sub-views in frequency. As seen in Table 2, E-BDL only increases parameters by 0.1 G and requires approximately twice the inference time per sample compared to ResNet-18.

For the class-wise evaluation, E-BDL-ResNet achieves the highest precision (0.990) and perfect sensitivity (1.000) for normal gait, surpassing BDL-ResNet’s precision of 0.977 and sensitivity of 0.996, and ResNet’s precision of 0.877 and sensitivity of 0.861. ConvNeXt-T also achieves perfect sensitivity (1.000) but with slightly lower precision (0.986). This indicates that E-BDL-ResNet can correctly identify normal gait patterns with minimal false positives. For the subtle abnormality group, E-BDL-ResNet significantly improves precision from ResNet-18’s 0.770 to 0.889 while maintaining competitive sensitivity (0.900). ConvNeXt-T has a slightly higher sensitivity (0.940) but lower precision (0.779). These results are desirable, indicating that the proposed method can better categorize normal walking and more accurately detect subtle abnormalities (early stage) with fewer “false alarms”. For the severe abnormality group, all candidates exhibit excellent performance. E-BDL-ResNet achieves perfect scores in both precision and sensitivity (1.000), marginally improving on BDL-ResNet’s precision of 0.995 and sensitivity of 0.997, and ResNet’s precision of 0.997 and sensitivity of 0.996. In the moderate abnormality class, E-BDL-ResNet shows competitive precision (0.889) and the highest sensitivity (0.867) compared to other models. ConvNeXt-T has a slightly higher precision (0.939) but lower sensitivity (0.725). This indicates that E-BDL-ResNet can maximally detect subjects in the moderate abnormality group with high sensitivity, allowing unhealthy subjects to be identified at an early step. E-BDL-ResNet balances precision and sensitivity well in detecting subtle and moderate abnormalities, but it is our intention to enhance the model’s ability to capture precise differences between subtle and moderate abnormalities.

The number of sub-band filters (*M*) is a hyperparameter in this approach and is determined based on validation results. As seen in Table 6, E-BDL-ResNet achieves the highest accuracy on both the validation and test sets with three sub-band filters. Therefore, *M* is set to three in this experiment. The learned parameters of the sub-band filters are displayed on the right side of Table 6. Using more sub-band filters aims to extract more sub-views from the input. However, using more filters does not necessarily lead to better performance. The accuracy results significantly decrease when the filter number is set to six. The centers of the filters are closer when the filter number is six, increasing the overlap among the filters. A large number of sub-filters, indicating more sub-views generated, challenges $SC_{Loss}$ in maintaining differences among salient sub-views, resulting in that $SC_{Loss}$ cannot guarantee a discriminative representation space for classification.

To further shed light upon the behavior of E-BDL-ResNet, the three learned filters are displayed in Figure 3a. Sub-views after filtering are shown in Figure 3c–e. Compared to the input spectrogram sample (Figure 3b), the second sub-view retains all frequency regions of the original sample, while the other two sub-views preserve distinct aspects of the sample by adjusting the power values in different frequency regions, respectively. Specifically, the first and third filters adjust the power values in the low-frequency regions (the middle part), while the first and third filters suppress the positive and negative high-frequency regions, respectively. The sub-view in Figure 3c mainly contains Doppler signatures from the torso, hips, legs, ankles, and feet, while Figure 3e emphasizes low-power signatures from the lower legs, ankles, and feet, allowing the low-power signatures to remain in the final representation vectors. This result highlights the interpretability of E-BDL. Additionally, the learning process of the sub-band filters is visualized in Figure A1. Compared to the sub-band filters at initialization (Epoch 0 in the figure), the first sub-band filter gradually adjusts the power values around 50–200 Hz, while the third one continuously reduces the power values around 0 to −130 Hz. The second sub-view tends to include more information from high-frequency ranges. This learning process demonstrates that the sub-band filters are able to adaptively identify salient sub-bands through gradient-based learning in an end-to-end manner.

In this study, we employed Grad-CAM++ [31] to generate heatmaps that visualize the attention regions of the backbone model on each sub-view. As mentioned above, the second filter retains the original input, while the first and third filters preserve distinct frequency regions. In Figure 4, distinct frequency regions are activated in each sub-view, indicating that the backbone model captures features from different salient sub-bands. Specifically, in Figure 4a, compared to the second sub-view, the backbone model focuses more on the regions above 100 Hz and below 64 Hz in the first and third sub-views, respectively. Similar effects can be observed in the subtle and severe abnormality examples. For the moderate abnormality, the backbone model is distracted by reverse leg movements and noise in the high-frequency regions. However, the third sub-view helps preserve features in the positive frequency regions. In general, relying solely on the second sub-view, which is equivalent to the original input, would cause the backbone model to miss some useful features. The first and third sub-views act as supplements to these lost features. Additionally, features detected from each sub-view implicitly embed the location information of frequencies. These characteristics ensure that E-BDL-ResNet effectively addresses the challenges posed by the band-dependent patterns.

## 4. Discussion

The proposed framework holds significant potential for recognizing gait abnormalities and classifying hemodynamics scenarios, based on experimental results. E-BDL enhances the representation learning of DL-based models by identifying significant sub-views in the adaptive sub-band filtering module and making use of band-specific features in the sub-view contrastive module. Experiments are conducted using two datasets including the vital-sign dataset, which includes separate heartbeat and respiration signals across various frequencies, and the gait dataset, comprising gait patterns reflecting body parts such as feet, legs, and hips across varied frequency bands.

Experimental results on the vital-sign dataset have demonstrated that E-BDL-ResNet achieves competitive performance in both subject-dependent and subject-independent settings. Compared to the recent CNN-based models, EfficientNetV2-S and ConvNeXt-T, E-BDL can enhance the performance of ResNet-18 with barely an increase in computational cost. Particularly, E-BDL-ResNet exhibits superior ability in hemodynamic scenario classification in the LOO validation. It demonstrates the potential to address challenges related to subject variability. Additionally, E-BDL-ResNet outperforms other candidates in overall accuracy and excels in a majority of class-wise evaluations in the in the ADRD risk evaluation using gait signatures, where more complicate band-dependent patterns can be observed compared to the vital-sign data. The computational cost of E-BDL-ResNet also qualified its characteristics of efficiency in implementation and prediction. Rather than uniformly dividing sub-bands, as in the BDL framework, E-BDL automatically identifies distinct and salient sub-bands as sub-views to maximally detect valuable features from gait signatures. Its superiority in interpretability is also justified by the learned filter parameters. The experimental results also reveal the potential of E-BDL-ResNet with fewer “false alarms” to healthy subjects and higher sensitivity to mild-impaired patients.

In future research, these limitations should be addressed. First, the adaptive sub-band filter module utilizes a modified Gaussian distribution function with two trainable parameters: location and scale. The shape parameter in the function could also be treated as a trainable parameter with constraints. Second, the sub-views are handled independently in the final two modules. Incorporating a feature fusion section could effectively integrate knowledge from all sub-views. Additionally, E-BDL is model-agnostic, and its performance was validated using the pretrained ResNet-18 as the backbone encoder for generalization. There is significant potential to apply alternative encoders to E-BDL. Considering the validated superiority of the proposed method, it is expected to be implemented in edge radar computing equipment for real-time detection.

## 5. Conclusions

This paper introduces a novel DL-based framework, E-BDL, to address the challenges posed by band-dependent patterns in radar sensing applications. E-BDL is end-to-end trained to explore and utilize band-dependent features in TFR. In E-BDL, an adaptive sub-band filtering module learns to identify salient sub-bands as sub-views from the original input and adjust the power scale in each sub-view, enabling more adequate band-dependent features to be extracted in the representation learning module. A sub-view contrastive module with a novel loss function is proposed to generate discriminative sub-view clusters for classification and ensure that each sub-view cluster focuses on different and valuable sub-views in the representation space. Experimental results demonstrate the effectiveness and robustness of E-BDL, showing superior performance in both hemodynamic scenario classification and ADRD risk evaluation, which rely on physiological and kinetic motions detected by micro-Doppler radar, respectively.

## Figures and Tables

**Figure 1 sensors-24-04620-f001:**
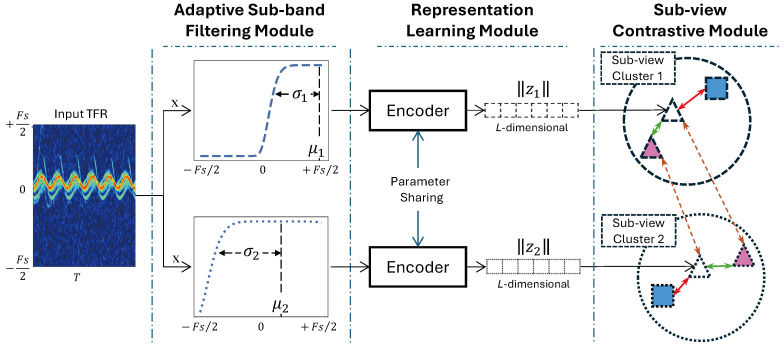
Illustration of E-BDL framework in the case of using two filters.

**Figure 2 sensors-24-04620-f002:**
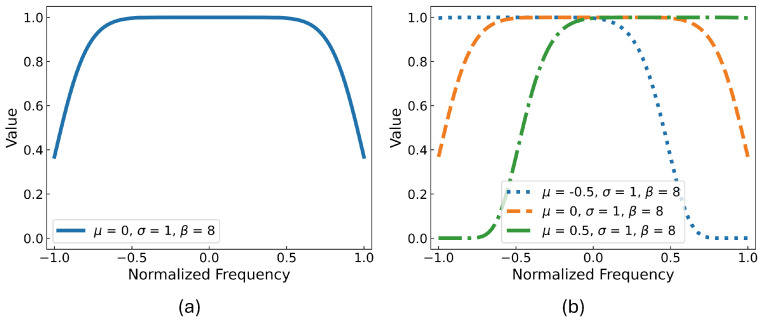
Illustration of adaptive sub−band filters. (**a**): A general example of a sub-band filter. (**b**): An initialization example while the number of filters is 3.

**Figure 3 sensors-24-04620-f003:**
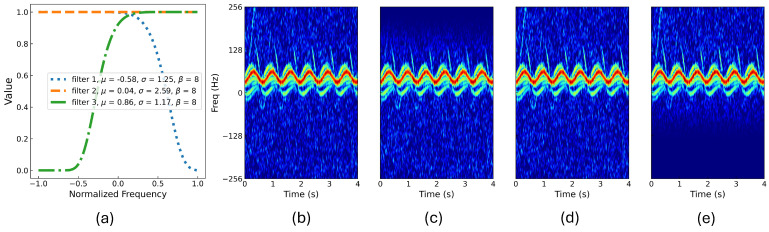
The learned filters and sub-views for the gait dataset: (**a**) The learned filters, (**b**) an example sample, (**c**) the first sub−view, (**d**) the second sub−view, and (**e**) the third sub−view.

**Figure 4 sensors-24-04620-f004:**
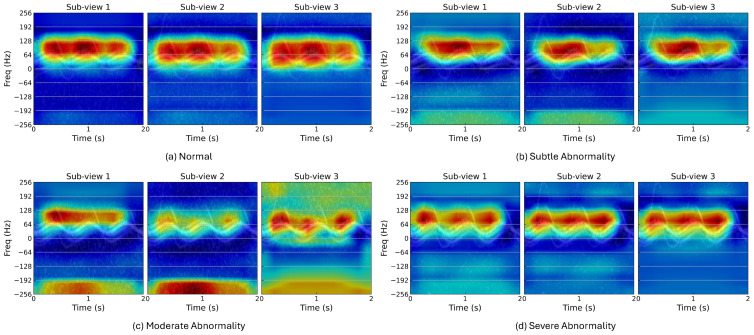
Heatmap visualization using Grad-CAM++.

**Table 1 sensors-24-04620-t001:** Summary of datasets for comparison.

Dataset	Frequency Band	# of Subjects	Categories	# of Samples
Vital-Sign	24 GHz	30	Resting	1860
			Apnea	152
			Valsalva	135
			Tilt-Up	1564
			Tilt-Down	1618
Gait	24 GHz	8000	Normal	2000
			Subtle Abnormality	2000
			Moderate Abnormality	2000
			Severe Abnormality	2000

**Table 2 sensors-24-04620-t002:** Summary of DL-based models for comparison.

Model	# of Parameters (G)	Inference Time per Sample (ms)	
		Vital-Sign Data	Gait Data
EfficientNetV2-S	20.8	10.4 ms	20.7 ms
ConvNeXt-T	28.2	6.2 ms	16.6 ms
ResNet-18	11.4	2.7 ms	3.3 ms
BDL-ResNet	11.4	-	3.1 ms
E-BDL-ResNet (M=3)	11.5	2.9 ms	6.3 ms

**Table 3 sensors-24-04620-t003:** Evaluation results on the vital-sign dataset (5-fold).

Model	Overall	Class-Wise									
		Resting		Valsalva		Apnea		Tilt-Up		Tilt-Down	
	Acc	Pre	Sen	Pre	Sen	Pre	Sen	Pre	Sen	Pre	Sen
ANN [11]	0.830	0.944	0.508	** 0.993 **	** 1.000 **	** 0.984 **	** 1.000 **	0.732	0.751	0.643	0.905
EfficientNetV2-S [26]	0.955	0.972	0.901	0.961	0.971	0.967	0.949	0.981	0.992	0.935	0.963
ConvNeXt-T [27]	** 0.969 **	** 0.973 **	0.934	0.977	0.978	0.974	0.964	** 0.985 **	** 0.997 **	0.942	** 0.970 **
ResNet-18 [19]	0.956	0.965	0.956	0.985	0.956	0.946	0.928	0.978	0.991	0.946	0.948
**E-BDL-ResNet**	0.959	0.970	** 0.967 **	0.943	0.963	0.972	0.914	0.984	0.992	** 0.959 **	0.959

**Table 4 sensors-24-04620-t004:** Evaluation results on the vital-sign dataset (LOO).

Model	Overall	Class-Wise									
		Resting		Valsalva		Apnea		Tilt-Up		Tilt-Down	
	Acc	Pre	Sen	Pre	Sen	Pre	Sen	Pre	Sen	Pre	Sen
EfficientNetV2-S [26]	0.714	0.552	** 0.502 **	0.887	0.873	0.793	0.877	0.805	0.874	** 0.476 **	0.446
ConvNeXt-T [27]	0.719	0.537	0.454	** 0.893 **	** 0.879 **	** 0.852 **	** 0.894 **	0.838	** 0.905 **	0.436	0.463
ResNet-18 [19]	0.707	0.558	0.439	0.874	0.862	0.826	0.862	0.853	0.872	0.463	0.550
**E-BDL-ResNet**	** 0.722 **	** 0.567 **	0.440	0.888	0.867	0.825	0.854	** 0.874 **	0.885	0.460	** 0.565 **

**Table 5 sensors-24-04620-t005:** Evaluation results on the gait dataset.

Model	Overall	Class-Wise							
		Normal		Subtle Abnormality		Moderate Abnormality		Severe Abnormality	
	Acc	Pre	Sen	Pre	Sen	Pre	Sen	Pre	Sen
EfficientNetV2-S [26]	0.900	0.949	0.997	0.774	0.888	0.925	0.718	0.998	0.994
ConvNeXt-T [27]	0.917	0.986	1.000	0.779	** 0.940 **	** 0.939 **	0.725	1.000	1.000
ResNet-18 [10]	0.884	0.877	0.861	0.770	0.876	0.899	0.805	0.997	0.996
BDL-ResNet [10]	0.923	0.977	0.996	0.822	0.902	0.920	0.794	0.995	0.997
**E-BDL-ResNet**	** 0.943 **	** 0.990 **	** 1.000 **	** 0.889 **	0.900	0.889	** 0.867 **	** 1.000 **	** 1.000 **

**Table 6 sensors-24-04620-t006:** Accuracy results of E-BDL-ResNet using different sub-band filter number.

Filter Number	Accuracy on Validation Set	Accuracy on Test Set	Learned Parameter Sets (μ, σ)
1	0.923	0.915	(−0.61, 3.13)
2	0.931	0.915	(−0.36, 2.43); (2.27, –0.56)
3	** 0.948 **	** 0.943 **	(−0.58, 1.25); (0.04, 2.59); (0.86, 1.17)
4	0.938	0.929	(−2.04, –0.65); (1.53, 2.36); (−0.67, 3.30); (0.93, 1.05)
5	0.940	0.936	(−0.66, 1.23); (−0.02, 2.70); (0.88, 1.73); (0.32, 3.35); (2.64, –1.07)
6	0.787	0.791	(−3.14, –2.63); (−2.66, –0.58); (−0.48, 3.52); (−0.82, 1.61); (0.60, 1.39); (0.75, 0.84)

## Data Availability

Dataset available on request from the authors.

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
