# Peer review of "E-BDL: Enhanced Band-Dependent Learning Framework for Augmented Radar Sensing"

_sensors, 2024, doi:10.3390/s24144620_

Round 1

Reviewer 1 Report

Comments and Suggestions for Authors

The authors propose an enhanced band-dependent learning (E-BDL) framework to effectively capture and exploit band-dependent features to improve classification in radar-based healthcare sensing applications. Comments are as follows:

1. This work is mainly based on the authors’ previously published article on BDL-ResNet, as described in [10]. Although [10] adopts simple frequency band division, this paper advances BDL-ResNet to E-BDL-ResNet by incorporating an adaptive sub-band filtering mechanism. Although the performance of frequency band division is improved, the computational load increases significantly due to adaptive sub-band division. Can the authors compare the computational load in the experimental section?

2. The authors claim that their sub-band division method is adaptive. Typically, adaptive methods utilize optimization functions or iterative processes to achieve adaptability. However, the method used in this paper does not demonstrate adaptive capabilities.

3. According to the reviewer's experience, training after separating different sub-bands is not as effective as directly extracting frequency features for training, because the deep neural network will automatically identify these features. In fact, the experimental results did not show significant improvement. The reviewer believes that sub-band division is a signal processing method and there are some alternatives in deep neural networks. The authors could also demonstrate the benefits of sub-band division from other perspectives besides accuracy.

4. The comparison method ANN is obsolete and does not need to be added. Comparison of the author's algorithm with ResNet does not show significant improvement, and some accuracy improvements appear to be within the error range. The authors are recommended to compare their method with deep neural networks from the past three years, even though they have not been used in the field of micro-Doppler radar, to better demonstrate the advancement of their work. If the accuracy of multiple algorithms is high, the author can appropriately increase the difficulty of identification to demonstrate the effectiveness of the proposed method.

5. The algorithm proposed by the author includes adaptive sub-band filtering, representation learning and sub-view comparison. Can its effectiveness be verified through ablation experiments?

6. Table 5 lists six filters but does not specify the parameters of the filter used in this example. Additionally, Figure 3 compares the performance of the three filters. Why not consider a bandpass filter? In theory, it should produce better results.

7. The authors propose to identify important sub-bands containing frequency band-related patterns as sub-views, while using a representation module to extract features and normalize them within each sub-view. How to ensure generalization performance through this approach should be discussed.

Comments on the Quality of English Language

The author's English is proficient.

Reviewer 2 Report

Comments and Suggestions for Authors

Dear Authors,

  The manuscript “E-BDL: Enhanced Band-Dependent Learning Framework for Augmented Radar Sensing” focuses on utilizing deep learning-based models for radar sensing applications, specifically exploring the characteristics of band-dependent patterns in radar data. You discussed the representation scheme for radar data, emphasizing the significance of time-frequency representation (TFR), particularly spectrograms, in signal analysis.

       The topic explored in your article is interesting mainly because of the characteristics of the images used, which is a challenging point. This issue could even be better explored in chapter one.

       The structure of the manuscript is adequate, but you could use more figures, for example, to better characterize the inputs used in the study.

    I have made some minor comments which can be found in the comments in the digital file. I would like to draw your attention to the following suggestions:

1) Improve the abstract: relate the challenge addressed to the proposed solution. Highlight the results obtained when compared to available solutions.

2) Lines 32 and 33: superiority and flexibility in which aspects? Transfer of knowledge?

3) Has Figure 1 been adapted from another reference? If so, please quote in the figure legend.

4) Lines 126 to 145: cite the bibliographical references that provide scientific support for the concepts and formula presented.

5) Line 197: comment further on the resulting probability distribution which allows the mean to be chosen rather than the median, for example.

6) Data preprocessing: were the parameters defined empirically or were they based on the theoretical framework?

7) Explore the large variation in accuracy seen when six filters are applied to the network.

8) The discussions can be deepened by comparing the results obtained in your experiment with those obtained by other authors and methods.

9) Revise the conclusions so that they respond adequately to the objectives proposed for the study.

I end my review by congratulating you on the study and the version of the manuscript you have submitted.

Respectfully,

Comments on the Quality of English Language

    It is necessary to review punctuation, especially the use of commas. Pay attention to long paragraphs, as they tend to confuse the reader. When acronyms are introduced for the first time, their meaning should be presented.

Round 2

Reviewer 1 Report

Comments and Suggestions for Authors

The authors have thoroughly addressed the reviewers' comments and made extensive revisions to the paper, which the reviewers now believe meets the publication standards of the Sensors journal.

Reviewer 2 Report

Comments and Suggestions for Authors

     Dear Authors,

    The manuscript “E-BDL: Enhanced Band-Dependent Learning Framework for Augmented Radar Sensing” focuses on utilizing deep learning-based models for radar sensing applications, specifically exploring the characteristics of band-dependent patterns in radar data.

     I'd like to congratulate you on the great work you have done in the implementation of the suggestions from the first round of revisions.

     After a detailed analysis of the new version of the manuscript, I can see that all the recommendations have been duly addressed. This has resulted in a significant improvement in the quality of the manuscript.

    I thank you for your commitment and dedication in revising and improving the manuscript. I trust that the changes you have implemented will make a positive contribution to the clarity and solidity of the manuscript presented.

    Thank you for sending me the cover letter to help me revise the new version of the manuscript.

     Respectfully,